# Short Alternative Route for Nuclear Fuel Reprocessing Based on Organic Phase Self-Splitting

**DOI:** 10.3390/molecules26206234

**Published:** 2021-10-15

**Authors:** Julie Durain, Damien Bourgeois, Murielle Bertrand, Daniel Meyer

**Affiliations:** 1ICSM, CEA, CNRS, ENSCM, University of Montpellier, Marcoule, CEDEX, 30207 Bagnols-sur-Cèze, France; julie.reaux@hotmail.fr (J.D.); daniel.meyer@cea.fr (D.M.); 2DMRC, CEA, BP 17171, CEDEX, 30207 Bagnols-sur-Cèze, France; murielle.bertrand@cea.fr

**Keywords:** solvent extraction, third phase, uranium, thorium, tributylphosphate (TBP)

## Abstract

A more sustainable management of natural resources and the establishment of processes allowing a joint management of nuclear materials to avoid their diversion from their civilian use are two issues for the nuclear industry. Short alternatives to existing processes have therefore been proposed based on known systems available, tributylphosphate (TBP), for the separation of actinides by liquid/liquid extraction. Proof of concept of such alternative has been established on the uranium(VI)/thorium(IV) system. From an organic phase consisting of a mixture of TBP/*n*-dodecane loaded with uranium and thorium, two fluxes have been obtained: the first contains almost all of the thorium in the presence of uranium in a controlled ratio, the second contains surplus uranium. Two levers were selected to control the spontaneous separation of the organic phase: the addition of concentrated nitric acid, or the temperature variation. Best results have been obtained using a temperature drop in the liquid/liquid extraction process, and variations in process conditions have been studied. Final metal recovery and solvent recycling have also been demonstrated, opening the door for further process development.

## 1. Introduction

Solvent extraction is one of the key technologies employed for separation and purification of metals [1]. Among its numerous applications, nuclear fuel reprocessing plays a central role in the development of a sustainable nuclear industry [2]. Pressurized water reactors (PWR) constitute the large majority of existing nuclear power plants, with the last generation of reactors—EPR, European Pressurized Reactor—being implemented today. These reactors use an enriched uranium-based fuel, composed of uranium oxide (UOX). Containing 3–5% of fissile ^235^U, this fuel generates fission products and plutonium [3]. France has long made the choice of reprocessing used fuel, in order to valorize both unburnt uranium and generated plutonium, through the preparation of fuel composed of mixed uranium and plutonium oxides—MOX, Mixed OXide fuel. Further developments anticipate the set-up of a next generation of reactors, fast neutrons reactors, which will rely on the use of rich plutonium MOX (up to 20% plutonium) [4]. The processes currently implemented at an industrial scale for the reprocessing of spent nuclear fuel involve five successive steps [5]: (i) the dissolution of the fuel allowing the solution of the elements, (ii) liquid/liquid extraction to separate the final waste and purify the elements of interest, eg., uranium and plutonium (PUREX process [6]), (iii) individual precipitation of both uranium and plutonium oxalates, (iv) calcination to obtain the corresponding oxides, and finally (v) mixing of the obtained powders, and shaping for preparation of new MOX fuel. These processes and the management of uranium-plutonium mixtures will have to evolve in order to comply with the increasing plutonium content. In addition, the nuclear industry continuously faces the risk of diversion of fissile material for non-civil purposes. Thus, any process development that would by-pass the un-necessary plutonium purification stages would bring an answer to the stakes faced by the nuclear industry nowadays. From this perspective, we propose a simple alternative process to the existing ones, still relying on solvent extraction with commercially available reagents that enable the management of a plutonium flux, which contains all plutonium, with a controlled amount of uranium (Figure 1). Such an approach where metals in a flux are not fully separated has already been investigated in the frame of nuclear fuel recycling, with processes such as the GANEX (Grouped ActiNides Extraction) process [7], based on the co-extraction of actinides and lanthanides, as well as in the frame of valuable metals recycling from waste [8].

During a solvent extraction (also called liquid/liquid extraction) process, the third phase formation phenomenon can be observed: the splitting of one of the phases causes the appearance of a new phase not present initially [9]. Such splitting can be observed on many systems, involving different solvents, leading to either a new organic phase, or to a new aqueous phase, or also to a micro emulsion [10]. The splitting of the organic phase has been the focus of extensive studies [11,12,13,14,15], as it has always been considered as an incident in nuclear extraction processes, due to changes in solvent hydrodynamics, criticality hazards in fissile materials, and loss of extraction performance. The aim of the studies was therefore to define the areas of existence of the third phase and ways of avoiding it by adding phase modifiers. However, organic phase splitting can be a major asset for the separation and concentration of solutes. Several examples have been reported, and the most successful one is a system consisting of phosphorus molecules and a hydrocarbon diluent allowing the recycling of the catalyst used in the synthesis of adiponitrile (a polyamide intermediate) during the DuPont process [16]. The mixture containing the organophosphite complexes of nickel(II) catalyzing the hydrocyanation reaction, the reagents (pentenenitrile), and the reaction products (dinitriles), are mixed with a light alkane (*n*-heptane or cyclohexane). After contact, self-splitting of the obtained organic phase results in a phase containing more than 85% of the phosphorus and metal complex molecules in the alkane and another phase containing reagents and hydrocyanation products [17]. The process was improved later on with similar organic phase splitting induced by a drop in temperature [18]. To the best of our knowledge, the technology is still employed in the production of adiponitrile, with an annual world capacity exceeding 1,000,000 metric tons.

Recently, we studied in detail the composition of organic phases arising from the self-splitting of organic phases containing U(VI) and Th(IV), composed of tributylphosphate (TBP) in *n*-dodecane [19]. An initial organic phase leads to two new organic phases: a heavy organic phase (HOP), that contains most of the TBP (at a concentration between 2 and 2.5 M), and a light organic phase, where TBP concentration is much lower (0.2 to 0.6 M), and referred to as dilute organic phase (DOP). We demonstrated that the heavy phase contains most of the Th(IV), whereas U(VI) distributes between heavy and light organic phases, and that the phenomenon is general, piloted by the concentration of Th(IV), and best explained by a supersaturation parameter, whatever the trigger of the splitting of the organic phase. In the present article, we detail how these results can be employed in order to propose alternative processes for spent nuclear fuel management, considering the elements given above, and employing Th(IV) as a surrogate for Pu(IV) because of the strong radioactivity of latter element. The goal was to accomplish a controlled separation of a classical organic phase mimicking the loaded solvent in the PUREX process (Figure 1), into two phases, the first one containing more than 95% of Th(IV) with a small amount of U(VI), in a ratio close to 1, and the second one containing the remaining U(VI). According to the initial U/Th ratio in the initial organic phase, lying around 10, this means that about 90% of the U(VI) will have to be found in the second organic phase. Two triggers to initiate the splitting of the organic phase have been studied: the addition of concentrated nitric acid, and the temperature drop.

## 2. Results

### 2.1. Nitric Acid Induced Phase Splitting

The first stage of the PUREX process is a counter-current extraction of both U(VI) and Pu(IV) from an aqueous HNO_3_ containing phase into an organic phase based on TBP 30 vol% in a mixture of dodecane branched isomers. The load in metallic cations in the organic phase depends principally on the volume ratio of organic to aqueous phases, when sufficient mixer-settlers are employed in order to completely extract all cations. As long as the total load in metallic cations remains lower than a critical concentration, called limiting organic concentration (LOC), the organic phase remains stable and does not split [20,21]. However, this LOC is strongly dependent on the aqueous HNO_3_ concentration, and it decreases with increasing HNO_3_ concentration in the aqueous phase [11,22,23]. As HNO_3_ is also extracted by TBP in the organic phase, increasing the aqueous HNO_3_ concentration also leads to an increase in organic HNO_3_ concentration. Thus, after extraction of metallic cations and phase separation, further addition of HNO_3_ in the organic phase should lead to the splitting of this phase (Figure 2). In order to validate this hypothesis, and characterize the distribution of metallic cations in the two newly generated organic phases, several organic phases were prepared following classical one-stage extraction procedure, and characterized (Table 1). These phases are all based on 1 M TBP in *n*-dodecane, close to the ca. 1.1 M TBP concentration of 30 vol% TBP, and enable us to study the effects of various parameters on the process (U/Th ratio, total concentration in metallic cations).

Upon addition of concentrated nitric acid in the organic phase, a spontaneous splitting occurs as long as the quantity of acid is sufficient to exceed the LOC. After addition of the nitric acid, the organic phase is gently homogenized with gentle shaking, and the two new organic phases rapidly segregate, and can be easily separated. As the LOC of the U/Th mixture depends mostly on the Th concentration, for a given total concentration of metallic cations in the organic phase (phases OP1, OP2, and OP3), the splitting occurs more easily at low U/Th ratio, and only a small volume of nitric acid is needed (3% respective to total organic phase volume, Figure 3). As expected, when splitting occurs, most Th is present in the Heavy phase (HOP), and U distributes between both phases. Whatever the initial U/Th ratio, the more concentrated HNO_3_ is added (7.5 vol% respective to organic phase volume), the higher is the amount of Th in the HOP. The amount of Th in the HOP exceeds 90%, except when the initial Th concentration is low (OP3). In all three cases, the final U/Th ratio in the HOP is more or less equal to the initial U/Th ratio in the organic phase (Table 2).

Upon increase of the total metallic cation concentration, with a similar U/Th ratio, the amount in Th present in the HOP increases at constant added concentrated HNO_3_ (Table 2, organic phases OP3 and OP4). Furthermore, at higher load in metallic cations (OP4), the splitting of the organic phase occurred at lower added HNO_3_ (only 3 vol%), with already 92% Th segregated in the HOP. This is in full agreement with our previous description of third phase formation in the case of U/Th mixture where we detailed that the higher the Th load in the organic phase, the higher the supersaturation degree, and the higher the proportion of Th found in the HOP [19]. However, the final U/Th ratio in the HOP is still close to the initial one: we examined various range of initial conditions (U/Th ratio) and phase splitting conditions (quantity of added concentrated nitric acid solution), and the final U/Th ratio in the HOP was found to be rather independent of the amount of added HNO_3_ (Table 2). This indicates that, in the process, only the amount of Th found in the HOP is controlled, and it will be complicated to control the U/Th ratio.

### 2.2. Temperature Drop Induced Phase Splitting

Temperature variation during the liquid–liquid extraction process was also examined. This parameter is generally understudied: only some experiments at temperatures higher than room temperature are performed in order to characterize thermodynamic outcome of the process (exo- or endo-thermic features) [24]. The LOC is also strongly dependent on temperature, and lowers upon heating; it is possible to perform a liquid–liquid extraction at high temperature without occurrence of the third phase. Then, upon cooling, the organic phase so obtained, highly concentrated in metallic cations, will split (Figure 4a). The LOC of U + Th mixtures have been previously determined, and it has been established [19] that they increase linearly with temperature, with a slope of 2.3 ± 0.3 mmol·L^−1^·K^−1^ that does not depend on the U/Th ratio in the organic phase, nor on other parameters such as TBP concentration or aqueous HNO_3_ concentration. Only the intercept varies. For a 1 M TBP in *n*-dodecane organic phase contacted to a 6 M HNO_3_ aqueous phase, the LOC of a 9/1 U/Th mixture according to temperature is depicted in Figure 4b. The trend of organic phase total load in U + Th after a one-stage liquid–liquid extraction is also depicted, as the extraction of both U(VI) and Th(IV) by TBP is exothermic [25]. It is therefore anticipated that at high temperature no third phase will appear, whereas at lower temperature the organic phase should split.

In a first experiment, 1 M TBP was contacted to a 6 M HNO_3_ aqueous phase containing U(VI) and Th(IV) at 70 °C, and the phases were separated. The concentration of extracted metals was determined, and the organic phase was cooled to either 20 °C, 10 °C, or 1 °C. Spontaneous phase splitting occurred, and the distribution of U and Th in the two obtained organic phases was determined (Table 3).

The lower the temperature chosen for organic phase splitting, the higher the amount of metallic cations found in the HOP. Furthermore, the amount of Th segregated in the HOP is very high, above 90% as soon as the chosen splitting temperature is below 10 °C. The final U/Th ratio in the HOP is stable, around 4.5. With these encouraging results, the impact of several parameters was investigated (Figure 5). Solvents prepared from 0.8 M–1.2 M TBP in *n*-dodecane were loaded at 70 °C with U(VI) and Th(IV) through extraction from an aqueous phase based on 4 M , 5 M, or 6 M HNO_3_. The U/Th ratio targeted in the organic phase before splitting was 9, and it was found to lie between 8.8 and 10.2. Final U/Th ratios in the HOP obtained after phase splitting are also depicted in Figure 5.

As in the first studied case (1.0 M TBP and 6 M aqueous HNO_3_) presented beforehand, in all tested conditions, the lower the splitting temperature, the higher the amount of U(VI) and Th(IV) found in the HOP. Overall, for the same TBP concentration, the percentages of actinides in HOP increase with the concentration of nitric acid. Separation performance is similar between 5 M and 6 M HNO_3_ with 1 M and 0.8 M TBP. Concerning the concentration of TBP, it is noted that the greater the concentration of TBP, the greater the percentage of U(VI) and Th(IV) in the HOP. An exception is that for 1.2 M TBP and 6 M HNO_3_, the percentages of actinides in DOP are the lowest of all the conditions tested. For this sample, the total concentration of actinides in the organic phase before splitting was slightly lower than in the other cases. As a result, the supersaturation of the organic phase before splitting is lower than in other cases. The U/Th ratios in the HOP obtained were between 2.9 and 5.5; the lowest values were obtained for the lowest TBP concentration of 0.8 M. The highest values were observed for 1 M and 1.2 M TBP for 5 M HNO_3_. For the other HNO_3_ concentrations, the U/Th ratios obtained for 1 M and 1.2 M TBP are close, with an average value of 4.4.

These experiments show that for experimental conditions varying around 1 M TBP and 6 M aqueous HNO_3_, interesting separations can be obtained as long as the supersaturation of the organic phase obtained at 70 °C is sufficient. However, a compromise must be made between the amount of Th(IV) segregated in the HOP and U/Th ratio in the HOP: when the quantity of Th(IV) segregated in HOP is maximized, this leads to a higher U/Th ratio. In the case of nuclear fuel recycling, it is preferable to target a high yield of Pu(IV) in the HOP, because the requirements lie in terms of plutonium recovery efficiency. In all cases, the two organic phases were easily separated, without any peculiar difficulty during phase disengagement. The final viscosities of HOP were found to lie between 4 and 16.5 mPa.s, and of DOP around 0.8 mPa.s, in comparison with 0.9 mPa.s for an initial 1 M TBP organic phase in *n*-dodecane.

### 2.3. Stripping of Metals and Recycling of Extraction Solvent

Results obtained in the case of the temperature drop induced separation being interesting, it is now important to verify how actinides in HOP and DOP can be recovered and whether solvent recycling can be considered in the case of organic phase splitting. The conditions for actinides back-extraction in the DOP and in the HOP have to be validated as these two phases have very different compositions. The TBP concentration in the DOP is expected to be between 0.2 M and 0.6 M, whereas in the HOP it is expected slightly above 2.0 M. Furthermore, the HNO_3_ content in the HOP is elevated, and HNO_3_ is back-extracted during the back-extraction stage, so that the aqueous phase is enriched in nitric acid. Therefore, it is much more difficult to back-extract HOP than DOP: TBP concentration is higher, and aqueous HNO_3_ concentration is higher; both parameters are known to increase both U(VI) and Th(IV) distribution coefficients. Additionally, in the experiments performed at the laboratory, the ratio of DOP and HOP volumes were determined with precision before taking an aliquot. This allows the recombination of both organic phases (DOP and HOP) after U(VI) and Th(IV) back-extraction in the same proportions, to obtain a final TBP concentration identical to the initial concentration. The complete process operated is summarized in Figure 6:

The complete process was performed using a 1 M TBP solution in *n*-dodecane, and starting from a 6 M aqueous HNO_3_ solution of actinides. After extraction of U(VI) and Th(IV) and organic phase splitting, a 3:1 ratio of DOP/HOP volumes was found. Both phases were aliquoted for analysis, and remaining organic phases were separately back-extracted using a 0.01 M aqueous HNO_3_ solution. As we were working in batch mode, a 10/1 aqueous/organic phase volume ratio was employed in a unique stage for DOP, and a 20/1 ratio for HOP (see Discussion section). In the case of HOP, as after back-extraction it remained yellowish, a second-back-extraction stage was performed using same conditions. During the first back-extraction stage, 85% U(VI) and 90% Th(IV) were revealed to have been extracted, and matter balance was complete after second back-extraction stage. Principal features of organic phases are given in Table 4 below:

Practically, further process development would rely on multi-stage counter-current processes, which are known to be much more efficient (see Discussion section). The recombination of DOP and HOP stripped from metallic cations was carried out with a 3:1 DOP/HOP ratio: 200 µL of stripped HOP were added to 600 µL of stripped DOP. The resulting organic phase was slightly turbid, and after a few minutes, a small water droplet separated, and the organic phase became clear. The new TBP based organic phase was then equilibrated with a 6 M aqueous HNO_3_ solution. The TBP concentration was determined after the total P content analysis with X-ray fluorescence (33.0 g/L P, ie. 1.06 M TBP), and extraction performances on U(VI) and Th(IV) extraction were found similar to those of a fresh 1 M TBP solution in *n*-dodecane.

## 3. Discussion

The behavior after phase splitting of the two actinides studied when they are in a mixture is different: Th(IV) has a strong affinity for the heavy phase (HOP) while U(VI) distributes between the heavy and diluted organic phases. This trend was exploited to propose separations to obtain a flow containing all the thorium in the presence of uranium and another flow of purified uranium. Organic phase splitting induced by acid addition or temperature variation has shown that it is possible to define conditions leading to interesting separation performance by obtaining more than 95% of thorium in heavy phase (HOP) with a U/Th ratio of the order of 4 for example. Best results have been obtained using a temperature drop induced phase splitting, and tests with different initial conditions and solvent recycling were carried out to validate this innovative separation concept. Latter route is preferred, also as the use of concentrated nitric acid solution is less practical. This strongly oxidizing reagent can react with organic molecules and lead to the appearance of so-called ‘red-oil’ during the process [26]. Furthermore, the thorium segregation in the HOP depends on the volume of added concentrated HNO_3_ solution, and when a too large volume was employed, a third aqueous phase appeared in the system. This aqueous phase is very concentrated in HNO_3_ and of little volume compared to both other organic phases. Back-extraction of metallic cations into this aqueous phase was not observed, but it may pose an issue in the organic phases separation process. The route induced by the drop in temperature seems to be much less technically restrictive while being effective regarding the segregation of thorium. The proposed separation is based on the successive production of two biphasic organic/aqueous and organic/organic systems. The sole requirement is the necessity to operate solvent extraction at 70 °C, followed by phase separation at same temperature, then another phase separation at a temperature close to 0 °C.

The validation of the concept using the temperature drop trigger was performed on small scale through batch experiments. Extraction and back-extraction stages were not optimized: it is clear that complete extraction of both U(VI) and Th(IV), as well as complete back-extraction of both metallic cations from the organic phases obtained, can be performed using pulsed columns or multi-stage counter current mixer-settlers. These apparatuses are currently operated at industrial scale at the La Hague facility in France, at ambient temperature. Operating the extraction sequence at 70 °C requires adaptation of the unit in order to keep both aqueous and organic phases warm. The key step of the proposed approach is the separation of the two newly generated organic phases, DOP and HOP, after temperature drop. Interestingly, the control of temperature is seldom considered in hydrometallurgy for the development of new processes. The extra costs associated with the need of a temperature control should be studied in detail and compared to the cost of long and complex extraction-scrubbing-stripping sequences. To obtain a total recovery of thorium in the HOP, a compromise must be made on the U/Th ratio in this phase, since the more thorium is present in the heavy phase, the more uranium is also present in this phase, which increases the U/Th ratio obtained. Flexibility in the choice of experimental conditions is an asset and offers opportunities to optimize such a process. Solvent recycling after combination of stripped DOP and HOP has been validated. This step requires knowledge of the ratio of DOP and HOP volumes after organic phase splitting, in order to obtain a solvent with the same TBP concentration as that initially used without having to add reagents (TBP or *n*-dodecane) at the head of operation. On larger scale, the separation of the two organic phases is a single stage operation, which does not rely on specific apparatus. The two separated phases can afterwards be handled at ambient temperature for classical back-extraction.

The concept was established on U(VI)/Th(IV) mixtures, but the final target remains U(VI)/Pu(IV) mixtures. The handling of plutonium requires specific facilities, and it is important to design experiments based on knowledge gained on Th(IV), as Th(IV) is generally considered as a first order surrogate of Pu(IV). The LOC of Pu(IV) in U(VI)/Pu(IV) mixtures has been previously studied, and it has been proven that the LOC of Pu(IV) decreases in the presence of U(VI) [27]. Based on these results, it is clear that Pu(IV) will induce the formation of the third phase as does Th(IV). Additionally, the maximum total load in metallic cations for a 1 M TBP phase is estimated at about 0.5 M. It has also been shown that the behavior of third phase formation with Pu(IV) is not linear according to aqueous HNO_3_ concentration, and much more sensitive to temperature than Th(IV) [11,28]. Thus, although process optimization will be needed, the temperature induced phase splitting is expecting to perform better in the presence of Pu(IV) than in the presence of Th(IV).

## 4. Materials and Methods

### 4.1. Chemical Reagents

TBP (97% purity) and *n*-dodecane (purity ≥ 99%) were purchased from Sigma-Aldrich and used without any further purification. Isopropanol (propan-2-ol, anhydrous, for analysis) was purchased from Carlo Erba Reagents, Val de Reuil, France. Nitric acid solutions were prepared using 69.5% concentrated acid purchased from Carlo Erba Reagents. Thorium(IV) nitrate hydrate Th(NO_3_)_4_·5H_2_O (99% purity) was purchased from Sigma-Aldrich, uranium(VI) nitrate hydrate UO_2_(NO_3_)_2_·6H_2_O was kindly provided by the CEA, France, and both were used without any further purification.

### 4.2. Experimental Procedures

Solvent extraction: TBP/*n*-dodecane solvent (TBP concentration varying between 0.8 M and 1.2 M) was pre-equilibrated through contacting with a five-fold volume of an aqueous HNO_3_ solution of the desired molarity (from 4 M to 6 M, same molarity as used with the metals during the extraction step). Equal volumes of pre-equilibrated TBP/*n*-dodecane solvent and aqueous HNO_3_ solution containing actinide(s) nitrate(s) were introduced in an Eppendorf tube and placed in a thermostated orbital mixer (Thermomixer^®^ C, Eppendorf, Montesson, France) set at the desired temperature (between 1 °C and 80 °C ± 1 °C). After thermal stabilization was reached, the tubes were vigorously shaken at 1000 rpm, and phases were allowed to set-up by gravity at same temperature in the same apparatus. Aliquots (between 25 µL and 100 µL) of final organic and aqueous phases were carefully taken and analysed (see below).

Nitric acid induced phase splitting: An organic phase (5 mL) was prepared at 20 °C as described above. Then aliquots of this phase (700 µL) were placed in Eppendorf tubes, and 10 to 50 µL of a concentrated HNO_3_ solution were added. The two new organic phases formed instantaneously, and were allowed to separate. Aliquots (25 µL for HOP and 100 µL for DOP) of both phases were carefully taken and analyzed (see below).

Temperature drop induced phase splitting: An organic phase (3 mL) was prepared at 70 °C as described above. Then aliquots of this phase (700 µL) were placed in Eppendorf tubes, and cooled separately to either 20 °C, 10 °C, or 1 °C in a thermostated orbital mixer (Eppendorf Thermomixer^®^ C) set at the desired temperature. The two new organic phases formed instantaneously, and were allowed to separate in the thermostated apparatus. Aliquots (25 µL for HOP and 100 µL for DOP) of both phases were carefully taken still when in the thermostated apparatus, and analyzed (see below).

### 4.3. Analysis Protocols

ICP/AES metal concentration determination: Aliquots of aqueous phases were directly diluted into a 2% HNO_3_ aqueous solution. Metals contained in the aliquots of the organic phases were back-extracted with 800 µL of an aqueous 0.01 M HNO_3_ solution at 20–22 °C for 1 h. Both phases were separated, and 500 µL of the resulting aqueous solution were taken and diluted into a 2% HNO_3_ aqueous solution. Concentrations of each metal in both aqueous phases (extraction and stripping phases) were determined by inductively coupled plasma atomic emission spectroscopy (ICP/AES, SPECTRO ARCOS ICP Spectrometer, AMETEK Materials Analysis, Elancourt, France). Given concentrations are calculated as the means of three replicates on three different wavelengths for each metal; relative standard deviations were determined and lie between 1 and 4%.

XRF TBP concentration determination: TBP concentration in organic phases was determined with an X-ray Fluorescence (XRF) spectrometer SPECTRO XEPOS apparatus (AMETEK Materials Analysis, Elancourt, France). An X-ray beam is generated from a Pd wire, and energy ranges are selected through reflection on a highly ordered pyrolytic graphite (HOPG) target under helium flush. The K-α line (E = 2.02 keV) of phosphorous was considered for quantification after calibration curves establishment. A volume of 100 µL of the organic phase of interest was diluted 20 times in isopropanol, and transferred into a 2.4 cm cuvette equipped with a 4 µm prolen^®^ foil and closed with a plastic cap. The cuvette was placed in the spectrometer. Molar TBP concentrations were calculated using [TBP] = [P]/M_P_ where [P] is the measured phosphorous mass concentration (in g/L) and M_P_ is the phosphorous molecular weight. Uncertainties on the concentrations were estimated to be 5% maximum.

## 5. Conclusions

Altogether, the performed study demonstrates the interest for the development of new separation techniques. The self-splitting of organic phases has been clearly underexploited so far, although it simply relies on temperature variation. In addition, the central idea of generating through a simple technique flows of incompletely purified metals requires deeper attention from the scientific community. The GANEX process is a rare example of such a development, albeit with a different objective, as the goal was to reduce the volume of generated nuclear waste through separation of unburnt uranium from activation and fission products (which encompass plutonium). Here, we propose to also recover plutonium for further valuation. Both approaches exploit the same idea of avoiding isolating pure plutonium. The technique proposed in the present study will also be transferred to other extraction solvents currently examined in the frame of spent nuclear waste reprocessing, such as monoamides and diamides. Additionally, the separation of other flows of metallic cations, such as those obtained in the frame of the recovery of valuable metals from wastes, will be investigated. In this case, combination of the self-splitting of organic phase followed by recovery of pure metal after selective precipitation, which is currently employed on single phases, could represent a real gain in process efficiency.

## Figures and Tables

**Figure 1 molecules-26-06234-f001:**
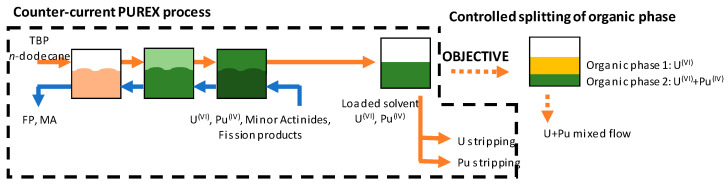
Schematic description of the first stage of the spent nuclear fuel reprocessing process (U and Pu extraction) and objective of the new proposed process.

**Figure 2 molecules-26-06234-f002:**
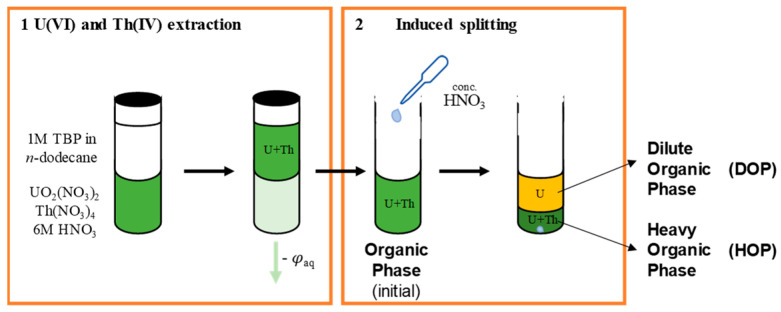
Schematic diagram of organic phase splitting induced by acid addition.

**Figure 3 molecules-26-06234-f003:**
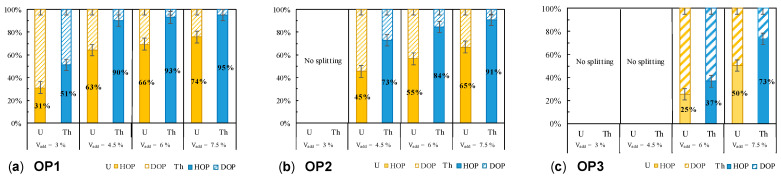
Distribution of U(VI) and Th(IV) in HOP and DOP after splitting of organic phases OP1 3 of low total U + Th concentration (64–68 mM) induced by addition of a controlled volume (Vadd) of concentrated nitric acid solution.

**Figure 4 molecules-26-06234-f004:**
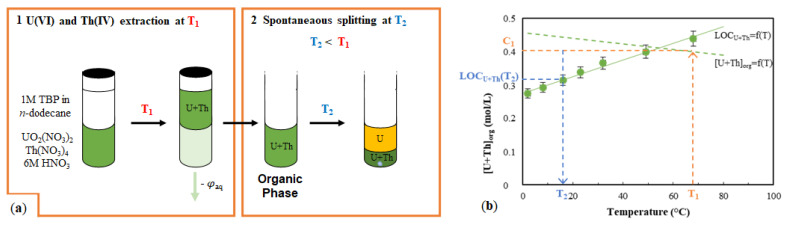
(**a**) Schematic diagram of organic phase splitting induced by temperature drop; (**b**) evolution of LOC and total extracted concentration of metallic ions in the organic phase according to temperature.

**Figure 5 molecules-26-06234-f005:**
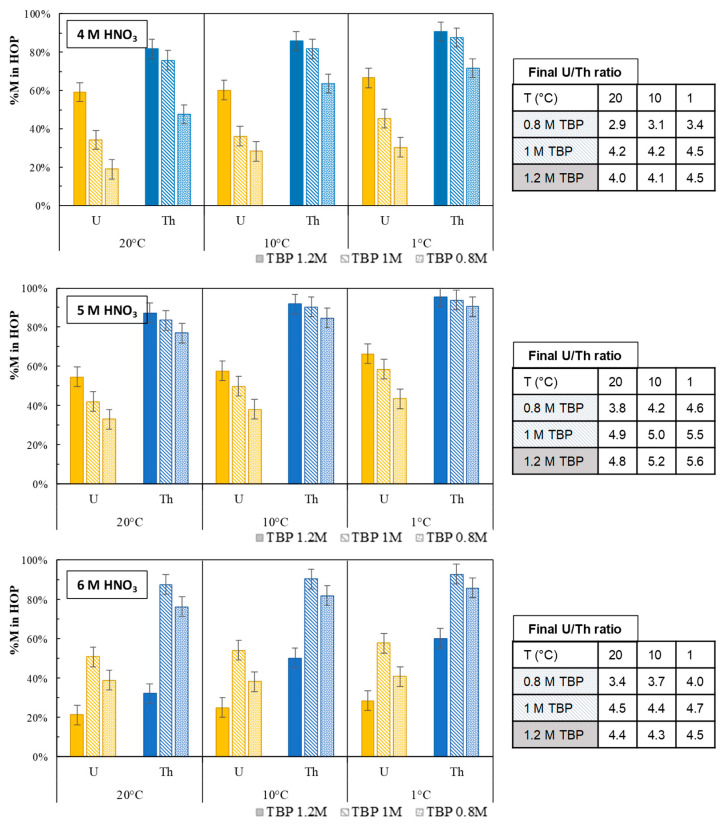
Distribution of U and Th in HOP and corresponding U/Th ratio after organic phase splitting, according to the aqueous HNO_3_ concentration, the splitting temperature (T, °C), and the organic TBP concentration.

**Figure 6 molecules-26-06234-f006:**
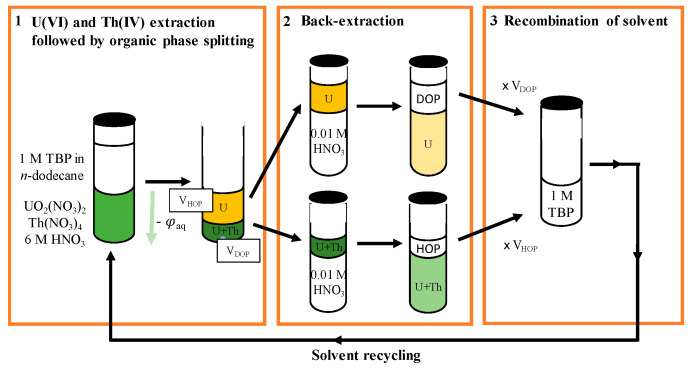
Schematic diagram of complete process, including solvent recycling, based on organic phase splitting induced by temperature drop.

**Table 1 molecules-26-06234-t001:** Organic phases employed in acid induced organic phase splitting. All phases are in 1 M TBP in *n*-dodecane, pre-equilibrated with 6 M aqueous HNO_3_.

Organic Phase	[U(VI)] (mM)	[Th(IV)] (mM)	[U] + [Th] (mM)	U/Th Ratio
OP1	8	60	68	0.13
OP2	35	27	65	1.3
OP3	58	5.85	64	9.9
OP4	302	28	330	10.8
OP5	173	32	205	5.4

**Table 2 molecules-26-06234-t002:** Detailed composition of heavy organic phases (HOP) arising from the splitting of organic phases OP1-5 induced by the addition of a variable quantity of a concentrated nitric acid solution.

Organic Phase	Initial U/Th Ratio	Quantity of Added HNO_3_ (vol%)	Final U/Th Ratio	% Th in HOP
OP1	0.13	3	0.08	51%
		4.5	0.09	90%
		6	0.10	93%
		7.5	0.10	95%
OP2	1.3	4.5	0.8	73%
		6	0.9	84%
		7.5	1.0	91%
OP3	9.9	4.5	(No phase splitting)	
		6	6.7	37%
		7.5	7.3	73%
OP4	10.8	3	7.2	92%
		4.5	8.2	95%
		6	8.7	97%
OP5	5.4	3	3.4	84%
		4.5	3.8	91%

**Table 3 molecules-26-06234-t003:** Outcome of organic phase splitting upon cooling to a give temperature of an organic phase prepared by contacting a 1 M TBP solution in *n*-dodecane to an aqueous 6 M HNO_3_ solution containing U(VI) and Th(IV) at 70 °C. Initial (U) = 395 mM and initial (Th) = 45 mM (U/Th ratio equal to 8.8) at 70 °C.

Temperature	% Th in HOP	% U in HOP	Final U/Th Ratio
20 °C	87%	45%	4.5
10 °C	90%	45%	4.4
1 °C	93%	49%	4.7

**Table 4 molecules-26-06234-t004:** Composition of organic phases obtained during the process depicted in Figure 6.

Organic Phase	Volume (mL)	[Th(IV)] (mM)	[U(VI)] (mM)	U/Th Ratio
Initial	1.6	88	364	4.1
Dilute (DOP)	1.2	3.5	152	43.4
Heavy (HOP)	0.4	342	1003	2.9

## Data Availability

Not applicable.

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
