# Peer review of "Short Alternative Route for Nuclear Fuel Reprocessing Based on Organic Phase Self-Splitting"

_molecules, 2021, doi:10.3390/molecules26206234_

Round 1

Reviewer 1 Report

The article focuses on the results of research that was focused on the development of a new process for separating of fissile components of nuclear fuel and recycling them back into the fuel cycle. This takes into account the strategy of reducing the probability of distribution of nuclear materials, notably the undesirability of deep purification of plutonium.

Questions and Comments:

Fig. 1 – correct “Counter-current current PUREX process”

“Figure 3. This is a figure. Schemes follow the same formatting.” Is this the actual name of the figure?

Were the authors able to qualitatively separate the formed HOP and DOP organic phases to determine uranium and thorium content?

L194: clarify the abbreviation POL

L51-55: The authors mention the GANEX (Grouped ActiNides Extraction) process. The process involves the extraction of uranium in the first cycle of the process by extraction, and monoamides are proposed to be used as an extractant. In section 3 - Discussion and Conclusion authors should detail likely advantages of the new actinide separation process over the GANEX process. In addition, it is also recommended to consider what technological features the process will have. For example, will new apparatuses be needed? Will the number of process steps increase?

Reviewer 2 Report

The paper “Short Alternative Route for Nuclear Fuel Reprocessing Based 2 on Organic Phase Self-Splitting”  presents some interesting and novel results, but in my opinion it can’t be published in this form.  The manuscript has to be revised thoroughly

  • Table 1:

OP3 U/Th ratio – there should be 9.7 not  9.9

  • Figure 3:
  • It can be the title (“This is a figure. Schemes follow the same formatting”). Please give the correct description of Figure 3
  • I try to find the relation between Table 1 and Figure 3 and I could not find it. Please add some clarification
  • In table 1 , the concentration is given in mM, and in Figure 3 the concentration is given in M. This is very confusing. Please correct that
  • Is it any correlation between Figure 3 and Table 2? If yes, give some indication in the text, please.

  • Table 3
  • according to my calculations made on the basis of what you wrote final U/Th ratio is:

20°C - 4.8 not 4.5

10°C – 5.2 not 4.4

1°C – 5.8 not 4.7

Please check it. If I am wrong, you will have to give a way how do you calculate these ratio. Otherwise correct it.

  • Table 4

Dilute (DOP) u/Th ratio should be 50 not 43.5

  • Line 252 it should be DOP/HOP ratio of 3:1
  • Line 271 It should be ….less practical.
  • I strongly recommend to separate discussion and conclusion I strongly recommend to separate discussion and conclusion. It is easier for readers to learn important points
  • Generale comment

Please check all calculations. There can be no questionable data in serious scientific work.
